# Similar sensorimotor transformations control balance during standing and walking

**Maarten Afschrift**[1]*, **Friedl De Groote**[2◉], **Ilse Jonkers**[2◉]

**1** Department of Mechanical Engineering, Robotics Core Lab of Flanders Make, KU Leuven, Belgium,
**2** Department of Movement Sciences, KU Leuven, Belgium

◉ These authors contributed equally to this work.
* maarten.afschrift@kuleuven.be

**Data Availability Statement:** All relevant data are within the manuscript and its Supporting information files.

**Funding:** MA was funded by Fonds Wetenschappelijke Onderzoek - Vlaanderen (FWO):

## Abstract

Standing and walking balance control in humans relies on the transformation of sensory information to motor commands that drive muscles. Here, we evaluated whether sensorimotor transformations underlying walking balance control can be described by task-level center of mass kinematics feedback similar to standing balance control. We found that delayed linear feedback of center of mass position and velocity, but not delayed linear feedback from ankle angles and angular velocities, can explain reactive ankle muscle activity and joint moments in response to perturbations of walking across protocols (discrete and continuous platform translations and discrete pelvis pushes). Feedback gains were modulated during the gait cycle and decreased with walking speed. Our results thus suggest that similar task-level variables, i.e. center of mass position and velocity, are controlled across standing and walking but that feedback gains are modulated during gait to accommodate changes in body configuration during the gait cycle and in stability with walking speed. These findings have important implications for modelling the neuromechanics of human balance control and for biomimetic control of wearable robotic devices. The feedback mechanisms we identified can be used to extend the current neuromechanical models that lack balance control mechanisms for the ankle joint. When using these models in the control of wearable robotic devices, we believe that this will facilitate shared control of balance between the user and the robotic device.

## Author summary

The stability of human standing and walking is remarkable, given that from a mechanical point of view standing and walking are highly unstable and therefore require well-coordinated control actions from the central nervous system. The nervous system continuously receives information on the state of the body through sensory inputs, which is processed to generate descending motor commands to the muscles. It remains, however, unclear how the central nervous system uses information from multiple sensors to control walking balance. In standing balance, such sensorimotor transformations have been studied. When standing balance is perturbed, previous studies suggest that the central nervous system estimates the movement of the whole body center of mass to activate muscles and

12ZP120N. (https://www.fwo.be/) The funders had no role in study design, data collection and analysis, decision to publish, or preparation of the manuscript.

**Competing interests:** The authors have declared that no competing interests exist.

control balance. Here, we investigated whether the same sensorimotor transformations underlie control of walking balance. We found that changes in muscle activity and ankle moments in response to perturbations of walking balance were indeed proportional to center of mass movement. These findings suggest that common processes underlie control of standing and walking balance. Our work is significant because it captures the result of complex underlying neural processes in a simple relation between the body's center of mass movement and corrective joint moments that can be implemented in the control of prostheses and exoskeletons to support balance control in a human-like manner.

## 1 Introduction

Most humans are extremely good at standing and walking without falling even in uncertain environments. This is remarkable, given the instability of the human skeletal system. Continuous adaptations of muscle activity are needed to control the relatively high position of the center of mass (COM) above a small base of support. This is achieved through sensorimotor transformations: the nervous system continuously receives sensory inputs, which are processed to generate descending motor commands to muscles [1]. These sensorimotor transformations cannot be explained by local reflexes alone but rely on supra-spinal processes that integrate sensory information from multiple sources to derive information relevant to the motor task, i.e. stabilizing the musculoskeletal system [2, 3]. Indeed, there is evidence that muscles are controlled by task-level feedback from COM kinematics while standing. Changes in muscle activity and joint moments in response to external mechanical perturbations of standing can be explained by delayed feedback from COM kinematics [4–6]. However, it is yet unclear whether COM kinematics feedback also captures sensorimotor transformations during walking.

Compared to standing, the sensorimotor transformations controlling balance during walking are less commonly studied. Prior studies of perturbed walking mainly described the postural strategies and found that subjects modulate muscle activity [7–10] to adjust foot placement (stepping strategy) [11] and the location of the center of pressure (COP) in the stance foot [8, 12]. Adjustments of COP location in the stance foot are mainly achieved through modulating the ankle moment and are therefore referred to as ankle strategy [8, 12, 13].

Following frontal plane perturbations, balance is mainly controlled using a stepping strategy. Medio-lateral foot placement is strongly correlated with COM position and velocity [11, 13–16]. Furthermore, reactive bi-lateral gluteus medius activity, which has an important contribution to foot placement, is also correlated with COM kinematics [14, 17]. These results suggest that COM kinematics are indeed important task-level variables driving the control of medio-lateral foot placement during perturbed walking [18].

Following sagittal plane perturbations, both stepping and ankle strategies are used to control balance. COM kinematics feedback can explain fore-aft foot placement [15]. However, the correlation between COM kinematics and foot placement is weaker in the sagittal than in the frontal plane [19], which can be attributed to the higher reliance on the ankle strategy. Indeed, when eliminating the ankle strategy through stilt walking, a strong correlation between COM kinematics and foot placement is observed in the sagittal plane [20].

It is currently unclear if, similar as in perturbed standing, delayed feedback of COM kinematics also underlies the ankle strategy in perturbed walking. Consistent with the finding that control of task-level feedback rather than joint-level feedback underlies reactive standing

balance [3], we expect that reactive ankle muscle activity and ankle joint moments during walking can be described by delayed linear feedback of COM position and velocity but not by delayed feedback from local joint angles and velocities. In contrast to standing, body configuration and intrinsic mechanical stability vary considerably during the gait cycle and with gait speed. As a result, the effect of a motor command (i.e. muscle excitation) on body movement strongly depends on the phase in the gait cycle ([10, 21]). We therefore expect that sensorimotor gains, describing the strength of the feedback relation between COM kinematics and muscle activity, are modulated during the gait cycle and with gait speed. First, it has been shown that local reflexes, assessed through H-reflexes, are modulated during the gait cycle [22] and with walking speed [23]. Second, reactive muscle activity changes when applying mechanical or sensory perturbations at different instances in the gait cycle [1, 7, 9, 10]. For example, large changes in reactive calf muscle activity are observed during mid-stance [9, 10] but not during early stance or swing [10]. It has however not been investigated whether COM kinematics feedback with phase-dependent gains can explain these changes in reactive muscle activity.

If COM kinematics feedback can explain the ankle strategy and is indeed modulated during the gait cycle, this raises the question which sensory mechanisms underlie this modulation. Cutaneous stimulation studies suggest that sensory information from tactile sensors in the foot modulates reflex gains during walking [24, 25]. We therefore investigated whether tactile sensors also modulate task-level feedback gains and thereby contribute to the observed modulation of reactive muscle activity.

In this study, we investigated the sensorimotor transformations underlying the ankle strategy to control balance during walking. We hypothesized that, comparable to standing, feedback from COM kinematics, and not local joint-level feedback, can explain reactive ankle muscle activity and ankle joint moments in perturbed walking. However, given the changes in the stability of the musculoskeletal system and changes in body kinematics during the gait cycle and with walking speed, we hypothesized that feedback gains will be modulated during the gait cycle and with gait speed based on input from tactile sensors. To test these hypotheses, we evaluated the relation between the inputs of the feedback law (delayed COM and ankle joint kinematics) and the resulting motor command (muscle activity and joint torques) in four available datasets of perturbed standing and walking at different walking speeds in young healthy adults. This linear feedback model is based on previous studies on standing balance [3] and was adapted to allow modulation of feedback gains during the gait cycle. The available datasets contained different perturbation modalities, i.e. support surface translations and pushes, which excited the (neuromechanical) system in multiple ways (Fig 1), thereby guaranteeing the generalizability of our findings.

## 2 Results

In short, we showed that (1) sensorimotor transformations underlying the ankle response to perturbations of standing and walking can largely be explained by task-level feedback of COM kinematics and not with delayed linear feedback of joint kinematics and (2) COM kinematics feedback gains are modulated within the gait cycle and with walking speed.

### 2.1 Model for task-level feedback of COM kinematics

We evaluated the relation between COM kinematics and ankle muscle activity and joint moment in perturbed standing and walking. Therefore, we tested whether reactive muscle activity and reactive joint moments could be explained by a linear combination of the delayed

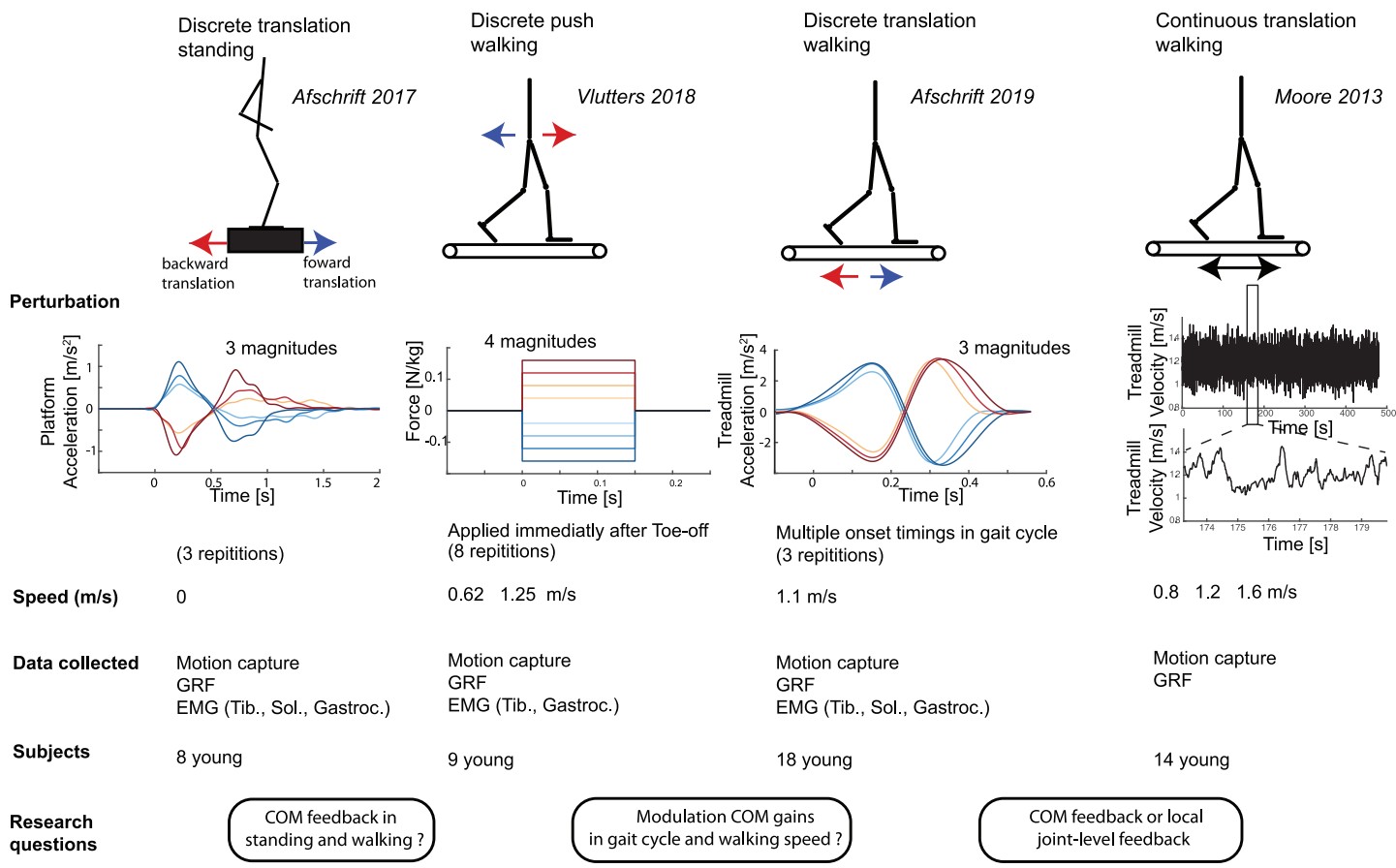

**Fig 1. Overview datasets and hypotheses.** We answered three research questions related to sensorimotor transformations underlying balance control by combining four previously published datasets with motion capture data of unperturbed and perturbed standing and walking. We used data of surface perturbation in standing [26] and walking [9, 10, 27] to evaluate if the ankle strategy is related to deviations in COM kinematics. Balance was perturbed during walking using both discrete and continuous surface perturbations [10, 27] and discrete pelvis push perturbations [9]. The potential modulation of feedback gains during the gait cycle was evaluated using a dataset with continuous surface perturbations during walking [27] and a dataset with perturbations applied at discrete instances of the gait cycle [10]. Finally, we used data with discrete surface translations in standing and walking to evaluate if altered sensorimotor transformations can explain the adjusted kinematic strategies to control balance observed in older adults. In all these experiments combined, joint kinematics and ground reaction forces were measured in 41 young subjects in total.

deviation of COM position and velocity from the unperturbed reference trajectories.

$$\Delta T_a(t) = K_p(s)\Delta COM(t - \tau_T) + K_v(s)\Delta C\dot{O}M(t - \tau_T) \tag{1}$$

$$\Delta EMG(t) = K_p(s)\Delta COM(t - \tau_m) + K_v(s)\Delta C\dot{O}M(t - \tau_m) \tag{2}$$

with $\Delta T_a$ the reactive ankle joint moment, $\Delta EMG(t)$ the reactive ankle muscle activity, $K_p(s)$ the position gain, $K_v(s)$ the velocity gain, and $\Delta COM$ and $\Delta C\dot{O}M$ the deviation of the the whole body COM position and velocity. Note that the feedback gains ($K_p(s)$ and $K_v(s)$) depend on the phase in the gait cycle ($s$). A neural delay of 60ms was used for muscle activity ($\tau_m$) and 100ms for joint moments ($\tau_T$). The delay in muscle activity is caused by the time needed for signal transmission and sensory integration in the nervous system [3], whereas the delay in joint moments is larger due to the additional electromechanical delay between muscle excitation and the development of force in the muscle. The sensitivity of the results to the time delay is discussed in Fig 7.

We compared the ability of the COM feedback model to explain experimental data with an alternative model based on delayed feedback of ankle angle and angular velocity.

$$\Delta T_a(t) = K_p(s)\Delta q_a(t - \tau_T) + K_v(s)\Delta \dot{q}_a(t - \tau_T) \tag{3}$$

$$\Delta EMG(t) = K_p(s)\Delta q_a(t - \tau_m) + K_v(s)\Delta \dot{q}_a(t - \tau_m) \tag{4}$$

with $\Delta q_a$ and $\Delta \dot{q}_a$ the deviation of the ankle joint angle and velocity from the unperturbed reference trajectory. A shorter neural delay of 40ms was used for muscle activity ($\tau_m$) and 80ms for joint moments to model the shorter latency of local reflex as compared to centrally mediated feedback pathways.

Feedback gains were estimated from the measured kinematics, joint moments and muscle activity by solving a least squares regression. Inputs, i.e., COM kinematics or ankle kinematics and outputs, i.e., ankle moments were selected respectively at 150ms and 230–250ms after perturbation onset since large deviations in COM kinematics were observed at this time instance for all different types of perturbations (see Methods section for details). Similarly, feedback gains for the ankle muscles were estimated based on COM kinematics and muscle activity respectively at 150ms and 210ms after perturbation onset. Feedback gains were estimated for each subject individually and with the data pooled over all subjects to evaluate if subjects use similar feedback gains. A visual representation of this method to estimate feedback gains can be found in Fig 2.

Uncentered coefficients of determination ($R^2$) and Root Mean Square Errors (RMSE) of the measured and reconstructed joint moments, as well as muscle activity are reported to quantify the fit of the linear models. Similar as in [28], all quantities and results are non-dimensionalized using COM height during quiet standing ($l_{max}$), the gravitational acceleration ($g$) and

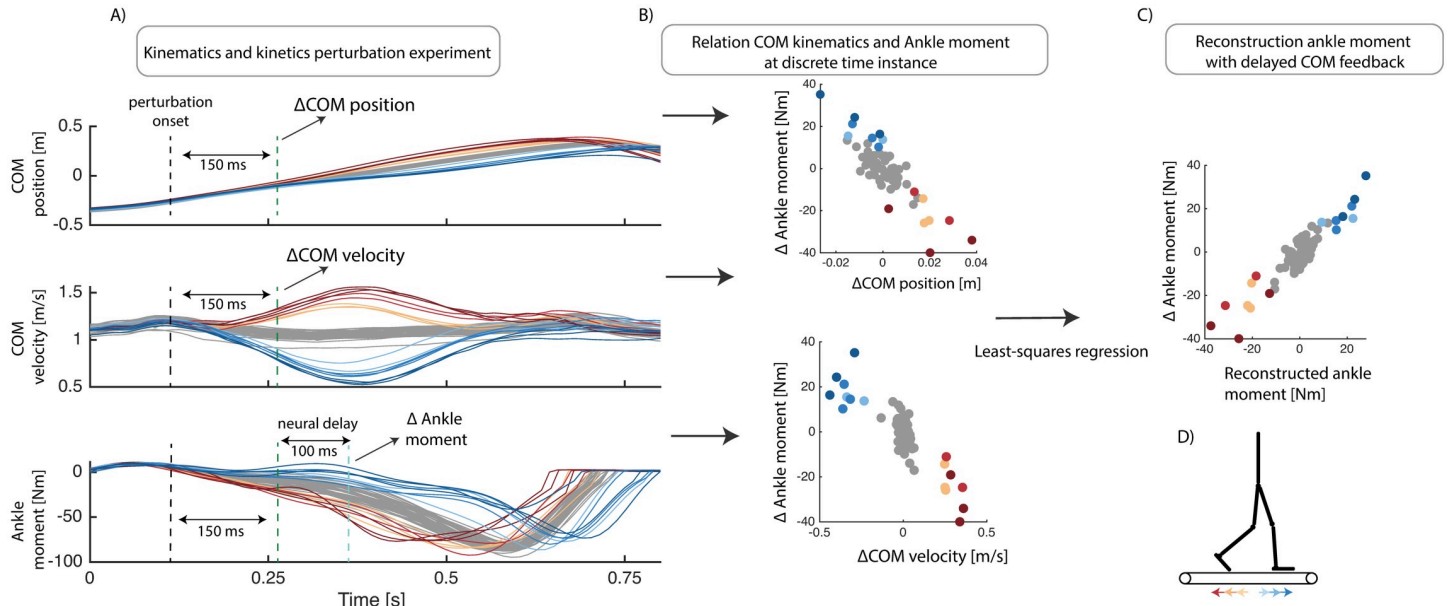

**Fig 2. Least-squares regression to evaluate the relation between delayed feedback of COM kinematics and ankle joint moment in reponse to belt-speed perturbations during walking (Data from [10]).** Deviations in COM position and velocity from unperturbed trajectory (gray), 150 ms after a belt speed perturbation, were used as input in a least-squares regression model. The deviation in ankle moment from the unperturbed trajectory, with an additional 100ms neural delay, was used as output in the regression model(B). The uncentered coefficients of determination ($R^2$) and Root Mean Square Errors (RMSE) of the measured and reconstructed joint moments are reported to quantify the fit of the linear models (C).

body mass ($m$). COM positions were normalized by $l_{max}$, speeds by $\sqrt{gl_{max}}$, torques by $mgl_{max}$ and muscle activity by maximal voluntary contraction values in the standing data and by maximal activity observed during unperturbed walking in the walking data (maximal voluntary contraction was not available in the walking data).

## 2.2 COM feedback explains ankle muscle activity and moment in perturbed standing

In agreement with previously published observation [5, 29], we found that reactive ankle joint moments and muscle activity during standing can be explained by delayed feedback of COM kinematics. Delayed feedback of COM position and velocity can explain the change in ankle joint moment ($R^2 = 0.94$, $RMSE$ 0.008) and muscle activity ($R^2$ between 0.62 and 0.89,) in response to forward and backward perturbations of different amplitudes (Fig 3 and Table 1).

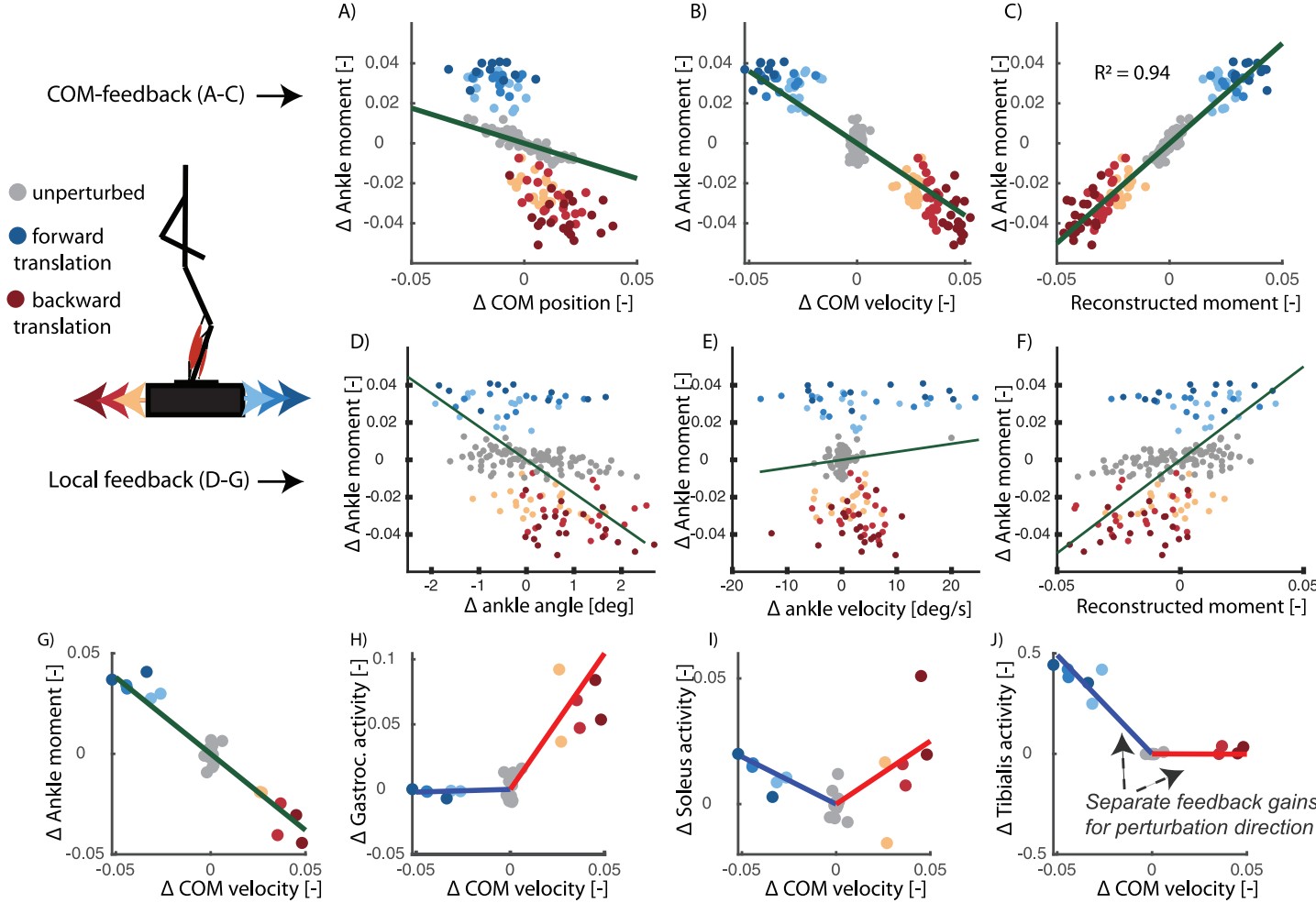

**Fig 3. Perturbed standing (data from [26]).** Here we show the relation between COM kinematics and ankle moment (A-C) and the relation between ankle kinematics and ankle moment (D-F) 150ms after perturbation onset. Each dot represents a single perturbation trial with a platform translation in forward (blue) or backward (red) direction with different magnitudes (color tint). A feedback model based on delayed COM position (A) and velocity (B) feedback can explain 94% of the variance in ankle moment (C) in response to support surface translations in forward and backward directions of different magnitudes. The linear feedback model based on angle angle (D) and angular velocity (E) explained only 21% of the variance in ankle moment (F). When analysing the data for each subject individually in the COM feedback model (G-J), we found that gastrocnemius and soleus activity increased with forward COM velocity and tibialis anterior activity increased with backward COM velocity. Each dot in graphs (G-J) represents a single perturbation trial of a selected subject. Note that the electromyography data was analysed for individual subjects instead of pooled over all subjects due to the limitations related to normalizing electromyography data.

**Table 1. Perturbed standing (data from [26]).** RMSE and $R^2$ values of the reconstruction of the ankle joint moment and calf muscle activity in perturbed standing with COM feedback or ankle joint kinematics feedback. Joint moments are reported pooled over all subjects (Pooled) and for individual subjects (Subj.) with the standard deviation over subjects (std). P-values are reported of the paired ttest that compares the model with COM feedback and the model with ankle joint kinematics feedback. The results for muscle activity are only reported for individuals and not pooled over all subjects due to the limitations related to normalizing electromyography data.

| | COM feedback | | Joint feedback | | p:$R^2$ | p:RMS |
|---|---|---|---|---|---|---|
| | $R^2 \pm$ (std) | RMS ± (std) | $R^2 \pm$ (std) | RMS ± (std) | | |
| Ankle moment (Pooled) | 0.94 | 0.008 | 0.21 | 0.028 | | |
| Ankle moment (Subj.) | 0.92 (0.07) | 0.008 (0.002) | 0.31 (0.15) | 0.027 (0.007) | **< 0.001** | **< 0.001** |
| Gastrocnemius | 0.62 (0.38) | 0.004 (0.004) | 0.55 (0.33) | 0.008 (0.005) | 0.12 | 0.11 |
| Soleus | 0.82 (0.18) | 0.011 (0.011) | 0.58 (0.28) | 0.018 (0.015) | **0.01** | **0.02** |
| Tibialis anterior | 0.89 (0.14) | 0.039 (0.031) | 0.58 (0.25) | 0.13 (0.11) | 0.08 | 0.12 |

Tibialis anterior activity increased proportional to COM position and velocity in response to a backward directed perturbation and gastrocnemius activity increased proportional to COM position and velocity in response to forward directed perturbations (Fig 3). Feedback gains for the ankle moment, $R^2$ and RMSE values were similar when computed for individual subjects or pooled over all subjects, indicating that different subjects use similar feedback gains (Table 1). Ankle moments reconstructed using COM kinematics were more similar to the measured data than ankle moments reconstructed using ankle joint kinematics (Table 1).

## 2.3 COM feedback explains ankle muscle activity and moment in perturbed walking

Similar as in standing balance, we found that reactive ankle joint moments and muscle activity can be explained better by delayed feedback of COM kinematics, than by delayed linear feedback from ankle kinematics, in perturbed walking. COM kinematics explained ankle joint moments and muscle activity across perturbation protocols.

First, we analysed the relation between COM kinematics, joint angles and ankle moments in a dataset with pelvis-push perturbations [9] applied at toe-off of the contra-lateral leg. We found that the reactive ankle moment and muscle activity after the perturbation could largely be reconstructed by delayed COM feedback (Fig 4 and Table 2). Tibialis anterior activity increased proportional to COM position and velocity in response to a backward directed perturbation (Fig 4I) and gastrocnemius activity increased proportional to COM position and velocity in response to forward directed perturbations (Fig 4J).

Second, we performed the same analysis in a dataset with a sudden increase or decrease in speed of the treadmill belts. To compare both perturbations modalities, we only considered belt-speed perturbations that were applied at the same time instant of the gait cycle as the pelvis push perturbations (i.e. around toe-off of the contra lateral leg). Similar as in the pelvis-push perturbations, we found that the reactive ankle moment after the perturbation could largely be reconstructed by delayed COM feedback ($R^2 = 0.61$) (Fig 5).

In both experiments, ankle moments reconstructed using COM kinematics were more similar to the measured data than ankle moments reconstructed using ankle kinematics (Table 2 and Fig 5).

## 2.4 Modulation COM feedback during the gait cycle and with walking speed

We found that the COM feedback gains are dependent on the phase in the gait cycle and are modulated with walking speed both when perturbations happen at discrete time instants and

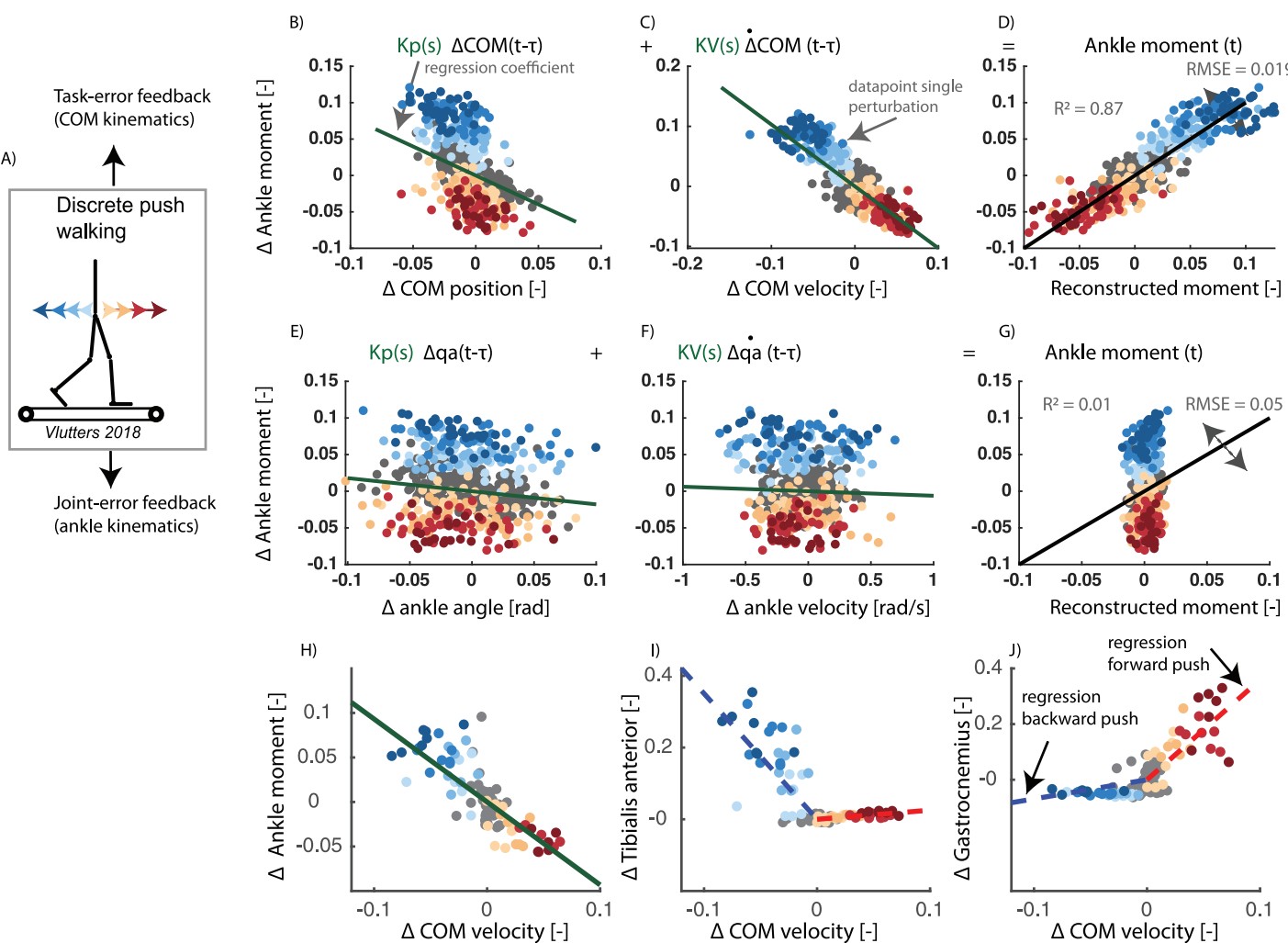

**Fig 4. Pelvis push perturbations walking (data from [9]).** Reactive ankle moment as a function of the deviation in COM position (A) and velocity (B) from the average reference trajectory during unperturbed walking (gray dots), in response to backward perturbations (blue dots) and forward perturbations (red dots). Least squares regression was used to estimate the position ($K_p$) and velocity ($K_v$) feedback gains, i.e. slope of green lines (A and B). The uncentered correlation coefficient ($R^2$) and RMSE is represented in pane D by plotting the measured ankle moment as a function of the reconstructed moment based on COM kinematics. In the same dataset, a similar relation between reactive ankle moment and deviation in ankle joint angle (E) and velocity (F) was evaluated. Ankle moments reconstructed using COM kinematics (D) were more similar to the measured data than ankle moments reconstructed using ankle kinematics (G). There strong correlation between delayed COM velocity and the ankle moment (H) is also reflected in changes in tibialis anterior activity (I) and gastrocnemius activity (J). tibialis anterior activity increases with negative COM velocity (i.e. blue regression line), and gastrocnemius activity increased with positive deviation in COM velocity (i.e. red regression line). Note that a representative example was selected for the muscle activity (H-J) rather than visualizing data pooled over all subjects, since this is only possible for the ankle moment and not for muscle activity due to the limitations related to normalization of electromyography data. This is data of walking at 0.6 m/s with the data pooled over all subjects.

continuously. The observed change in feedback gains in the gait cycle in combination with the high $R^2$ values during mid-stance(30% of stride) and low $R^2$ values during initial (10% of stride) and terminal stance (50% of stride) (Fig 5) indicates that the control of COM kinematics (1) changes during the gait cycles, (2) is most pronounced during mid-stance and (3) is only active when the foot is in contact with ground.

First, we used a dataset with changes in the belt speed in anterior and posterior direction at discrete instances in the gait cycle [10]. At the level of ankle joint moments, estimated feedback gains ($K_p$ and $K_v$) depended on the phase in the gait cycle (Fig 6: all subjects, Fig 5: one representative subject). The position and velocity gains are highest during mid-stance and lowest

**Table 2. Pelvis push perturbations walking (data from [9]).** RMSE and $R^2$ of the linear regression between delayed COM kinematics and reactive ankle muscle activity in perturbed walking. Joint moments are reported pooled over all subjects (Pooled) and for individual subjects (Subj.) with the standard deviation over subjects (std). P-values are reported of the paired ttest that compares the model with COM feedback and the model with ankle joint kinematics feedback. The results of muscle activity are only reported for individuals and not pooled over all subjects due to the limitations related to normalizing electromyography data.

| | COM-feedback | | Joint-feedback | | p-$R^2$ | p-RMS |
|---|---|---|---|---|---|---|
| | $R^2 \pm$ std | RMS $\pm$ std | $R^2 \pm$ std | RMS $\pm$ std | | |
| **walking 0.62 m/s** | | | | | | |
| Ankle moment (Pooled) | 0.87 | 0.02 | 0.01 | 0.05 | | |
| Ankle moment (Subj.) | 0.88 (0.03) | 0.02 (0.003) | 0.23 (0.21) | 0.05 (0.008) | **< 0.001** | **< 0.001** |
| Gastrocnemius | 0.61 (0.21) | 0.07 (0.04) | 0.58 (0.18) | 0.07 (0.04) | 0.61 | 0.83 |
| Tibialis anterior | 0.73 (0.11) | 0.02 (0.01) | 0.50 (0.09) | 0.03 (0.01) | **0.004** | **0.004** |
| **walking 1.25 m/s** | | | | | | |
| Ankle moment (Pooled) | 0.84 | 0.02 | 0.33 | 0.04 | | |
| Ankle moment (Subj.) | 0.85 (0.08) | 0.02 (0.01) | 0.50 (0.29) | 0.03 (0.012) | **0.003** | **0.006** |
| Gastrocnemius | 0.65 (0.21) | 0.05 (0.02) | 0.56 (0.18) | 0.06 (0.02) | 0.17 | 0.19 |
| Tibialis anterior | 0.68 (0.14) | 0.02 (0.01) | 0.45 (0.20) | 0.03 (0.02) | **0.002** | **0.05** |

during the swing phase. The variance in ankle moment explained by the COM feedback model ($R^2$ values) is highest during mid-stance (Fig 5). A similar modulation is observed at the level of reactive ankle muscle activity (Fig 6). The proportional increase in Gastrocnemius and Soleus muscle activity with deviations in COM kinematics in response to forward perturbations was highest during mid-stance (Fig 6). The proportional increase in tibialis anterior

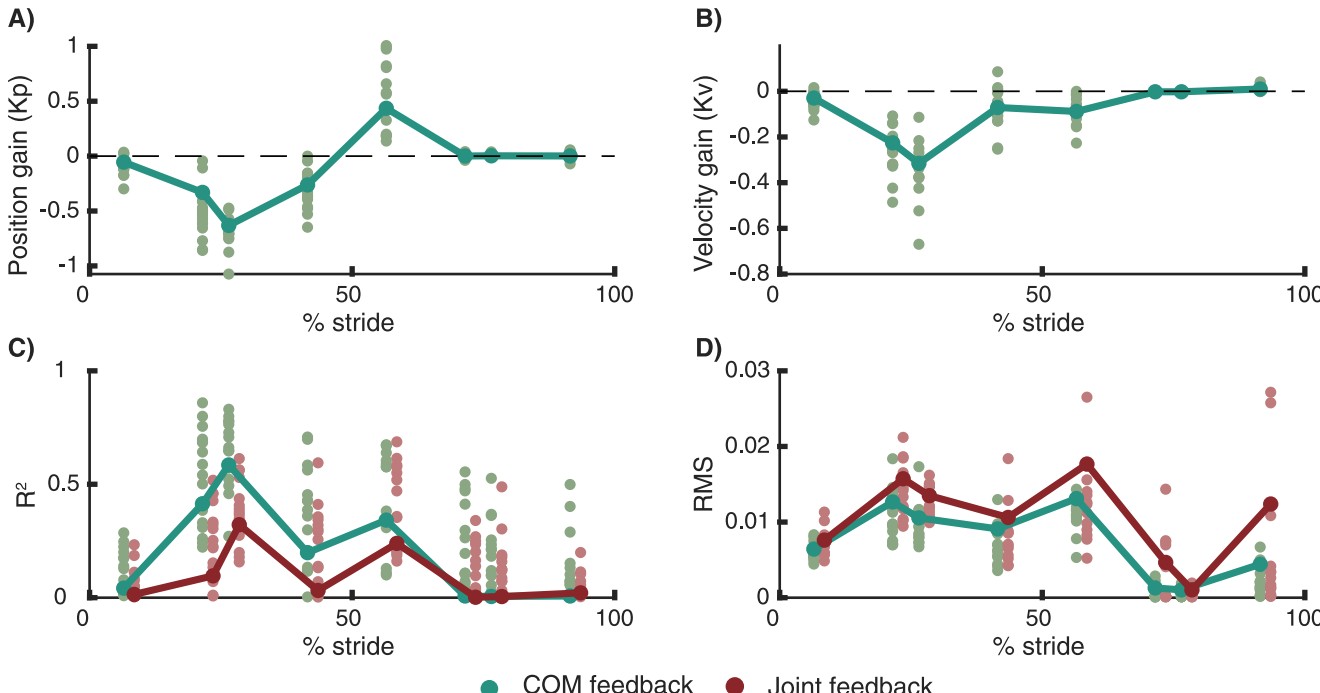

**Fig 5. Perturbed walking (data from [10]).** Gain modulation in response to beltspeed perturbations applied at different phases in the gait cycle. Reconstruction of the reactive ankle moments is better in a model with delayed COM feedback (green) compared to local joint level feedback (R) (C,D). The proportional feedback of COM position (A) and COM velocity (B) is modulated during the gait cycle. The bars and large dots represent feedback gains estimated based on the pooled data over all subjects. The small dots represent gains estimation in individual subjects. The perturbation onset timing in the gait cycle had a significant influence on the position gains, velocity gains, $R^2$ and RMS values.

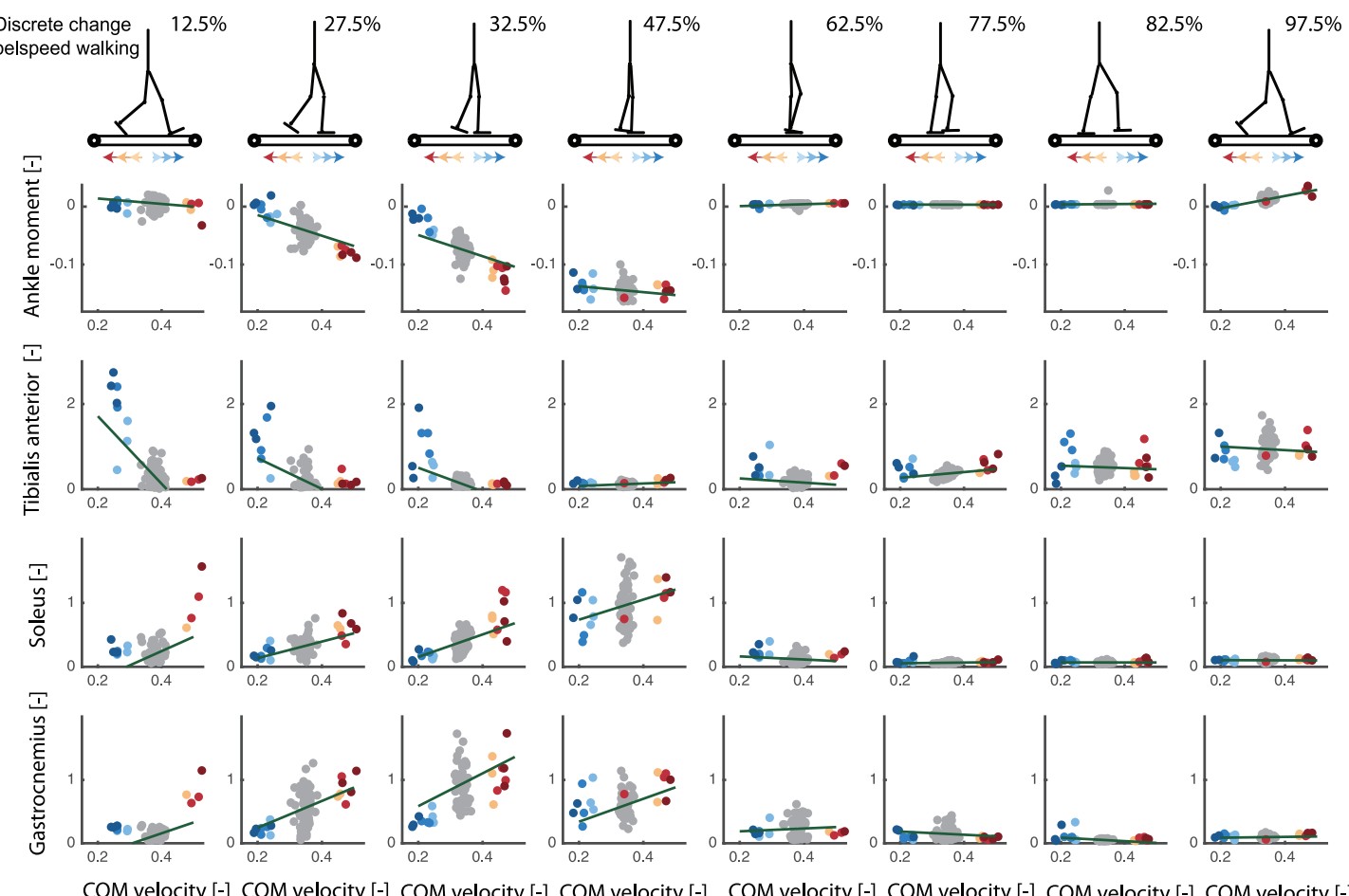

**Fig 6. Discrete treadmill perturbations walking (data from [17]).** Representative example of the relation between deviations in delayed center of mass velocity and reactive joint moments (row 1) and activity of the tibialis anterior (row 2), soleus (row 3) and gastrocnemius (row 4) in response to perturbations during walking. Perturbations were applied at four instances during the stance phase of the left leg resulting in eight responses at joint level when combining data of the left and right leg (e.g. right leg is in swing during mid-stance of the left leg). Unperturbed walking is visualised in gray, increases in belt speed (i.e. forward fall) in red and decrease in belt speed in blue (i.e. backward fall). Note that the total joint moment and muscle activity rather than the deviation from the average unperturbed data is shown here to also provide information on the muscle activity and joint moments during unperturbed walking.

activity with deviations in COM kinematics in response to backward perturbations was most pronounced during the first half of the stance phase. Remarkably, during mid-stance, calf muscle activity was also inhibited proportional to backward displacement and velocity of the COM in backward directed perturbations (Fig 6). This indicates that both muscle excitation and inhibition are proportional to deviations in COM kinematics in this phase of the gait cycle.

Second, we used a dataset with continuous changes in the speed of both belts while walking at 0.8, 1.2, and 1.6 m/s [27]. The main advantage of this data is that approximately 400 strides of perturbed walking could be analysed for each subject at multiple walking speed. To investigate phase-dependency of the reactive ankle moment within the stance phase, we divided the stance phase in 16 bins and computed a linear model for delayed COM feedback to ankle joint moment in each of those bins (Fig 7). Similar as in the discrete perturbations, we found that the changes in ankle joint moments during the gait cycle are closely related to COM position and velocity ($R^2$ up to 0.7 during mid-stance, Fig 8). Similar to the discrete perturbations, the $R^2$ values and feedback gains are low during early and late stance and high during mid-stance

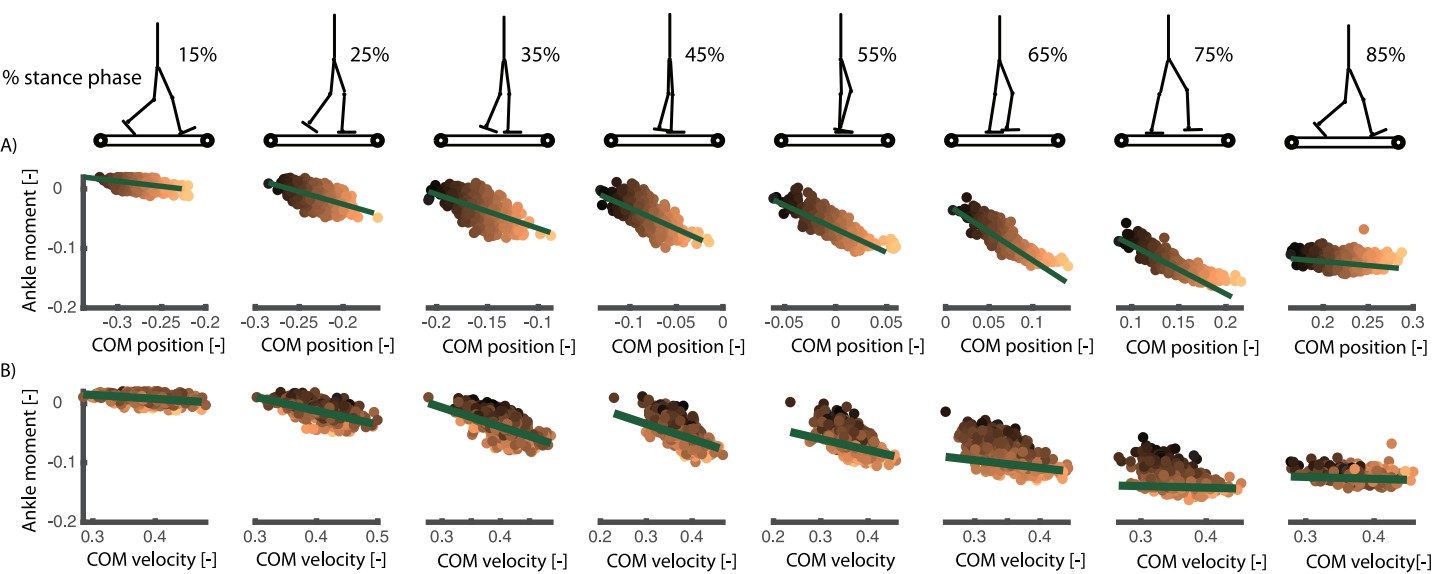

**Fig 7. Continuous treadmill perturbations walking (data from [27]).** Representative example of the relation between ankle joint moment and COM position and velocity in walking with continuous changes in the speed of both belts. The different columns in the plots are time bins equally spaced during the stance phase of walking. The colored dots represent the different strides of this subjects (with the color tint representing the deviation in COM position), the slopes of the green green lines are the resulting position and velocity gains from the least squares regressions. The change in slope of the green lines show the phase-dependency of the linear regression between COM position and velocity and the ankle moment during perturbed walking. Note that we show the total joint moment and muscle activity and not only the deviation from the average unperturbed data in each gait phase. This to indicate if a specific muscle is active, or the magnitude of a joint moment, in the gait phase during unperturbed walking.

(Fig 8). In addition, we found that feedback gains and the variance explained ($R^2$) by the linear regression decreased with increasing walking speed (Fig 8). Note that also for this dataset, the reconstructed ankle moment is more similar to the measured data when using COM feedback than using joint kinematics feedback for walking at slow and medium speeds(p-$R^2$ = 0.005 and

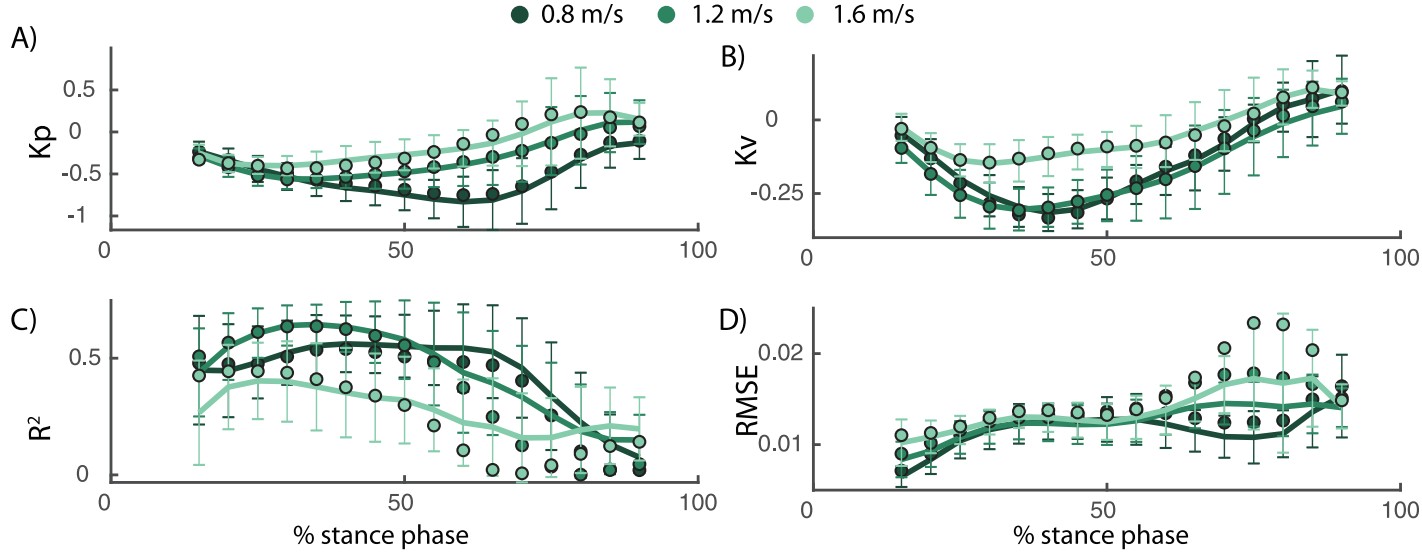

**Fig 8. Continuous treadmill perturbations walking (data from [27]).** The COM position (A) and velocity (B) feedback gains describing the reactive ankle joint moment depend on the walking speed. Both the position and velocity gains decrease with increasing walking speed. $R^2$ values decrease and RMSE increase with increasing walking speed, indicating a worse fit of the feedback model as walking speed increases. The errorbars and lines represent the average and standard deviation of the fit for individual subjects, the dots represent the results from the analysis based on pooled data over all subjects.

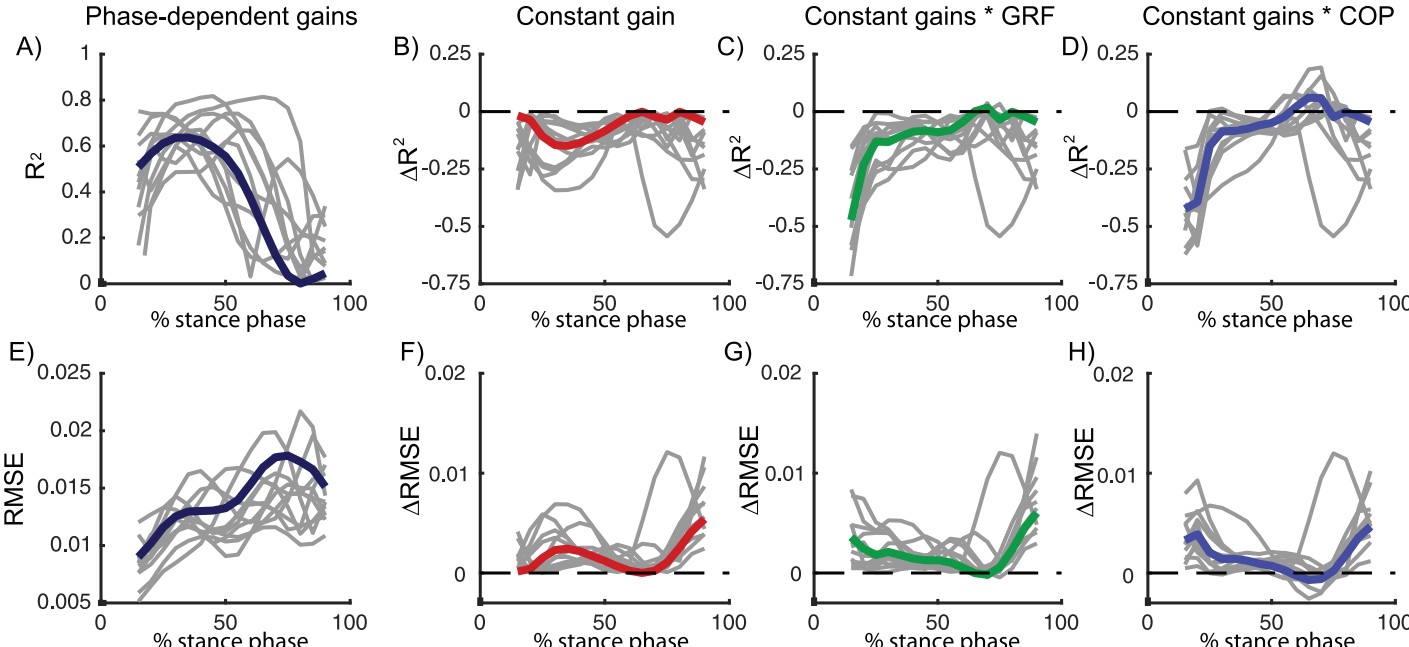

**Fig 9. Continuous treadmill perturbations walking (data from [27]).** $R^2$ and RMSE of the linear regression between COM kinematics and ankle moment in a model with phase-dependent feedback gains (A, dark blue). Relative to the model with phase-dependent gains, $R^2$ and RMSE values increased in models with constant feedback gains (B, red), constant feedback gains multiplied by vertical ground reaction force (C, green) and constant feedback gains multiplied by the COP position (D, blue). Data pooled over all subjects are visualised in color and results for individual subjects are visualised in gray. A similar increase in RMSE and decrease in $R^2$ was found in the three models, indicating that the quality of fit did not increase with information from the vertical force or COP. Results of the statistical analysis can be found in S1 Table in Fig 7.

p-$RMS$ < 0.001 for walking at 0.8 $m/s$ and 1.2 $m/s$) but not for walking at faster speeds (p-$R^2$ = 0.75 and p-$RMS$ = 0.59). The average $R^2$ and RMS values during the stance phase are respectively 0.38 and 0.014 with COM feedback and 0.23 and 0.018 with ankle kinematics feedback. The feedback model was not evaluated at muscle level since no electromyography data was collected in this dataset.

To evaluate if the modulation of feedback gains is important to describe the ankle strategy during the gait cycle, we compared the model with phase-dependent gains with a model with constant gains during the gait cycle (i.e. $K_p$ and $K_v$ are constant 1). For this analysis, we used the dataset with continuous treadmill perturbations during walking as this dataset contained perturbation data of many steps throughout the entire gait cycle. Due to the lower number of variables, we found that the $RMSE$ increased in the model with constant gains compared to the model with variable gains, especially at the end of the stance phase (Fig 9F). Similarly, the $R^2$ value decreased in the model with constant gains, especially during the first part of the stance phase. However, the average decrease in $R^2$ values (0.08) and in RMSE (0.002) in the constant gain model compared to the phase dependent gain model was small (Fig 9B and 9F).

## 2.5 Modulation COM feedback during gait cycle cannot be fully explained by tactile information from the foot

We hypothesized that the observed modulation of feedback gains during the gait cycle might be driven by cutaneous information from the foot. To test this hypothesis, we modified the constant gain model to include delayed feedback from cutaneous sensory information represented by either the location of the COP in the foot or the vertical ground reaction force. To

evaluate whether this model could explain the phase-dependent modulation of COM feedback, we used both a model in which the constant COM feedback gains were multiplied by delayed feedback from the vertical ground reaction force ($F_y$) or COP location ($COP_b$, minimal distance between the COP and the bound of the foot).

$$Ta(t) = Fy(t - \tau)(K_p \Delta COM(t - \tau) + K_v \Delta C\dot{O}M(t - \tau)) \tag{5}$$

$$Ta(t) = COP_b(t - \tau)(K_p \Delta COM(t - \tau) + K_v \Delta C\dot{O}M(t - \tau)) \tag{6}$$

Modulation of feedback gains by the vertical ground reaction force or COP did not improve the fit ($R^2$ and RMSE) compared to the model with constant gains (Fig 9C, 9D, 9G and 9H). In addition, for walking at 1.2 m/s, the modulation of constant feedback gains with the vertical ground reaction force resulted even in a significant decrease in the $R^2$ values compared to the model with constant gains (p = 0.01, see Table in S1 Table for details).

## 3 Discussion

### 3.1 Common sensorimotor transformations during standing and walking

We showed that common sensorimotor transformations can explain the ankle response in perturbed standing and walking. Both reactive ankle muscle activity and joint moments can be reconstructed by a linear combination of delayed COM position and velocity in perturbed standing and walking, rather than with linear feedback from ankle kinematics. Similar as in standing balance [3], we showed that the reactive EMG activity and ankle moments was better described by delayed feedback of COM kinematics across walking speeds and different perturbation modalities, compared to delayed feedback of ankle kinematics. This result supports the hypothesis that the nervous system estimates relevant task-level variables from multi-sensory information to activate muscles to control balance in response to perturbations [6, 18]. The neural mechanisms behind the task-level feedback of COM kinematics remains however unclear. Sensory perturbation experiments, with for example galvanic stimulation, might provide more insight in neural integration of sensory information from multiple sources to estimate COM kinematics.

### 3.2 Modulation feedback control during walking

We hypothesized that task-level feedback from COM kinematics are modulated during walking to compensate for changes in body configuration during the gait cycle. We indeed found that estimated feedback gains changed over the gait cycle in two datasets with perturbations during different phases of the gait cycle [10, 27]. Both the feedback gains and the variance in ankle moment explained by the linear regression were high during mid-stance but low during initial and terminal stance and during swing. Hence, the ankle strategy is mainly used during mid stance, which is logical from a mechanical point of view. In the ankle strategy, balance is controlled by modulating the interaction with the ground [8]. More specifically, activity of the ankle muscles changes the COP position in the foot to control balance [8, 10]. This mechanism has a high potential to control COM movement during mid-stance since the COP is approximately in the middle of the foot and can move in forward and backward directions. Hence, our results indicate that the proportional feedback of COM kinematics is modulated to exploit this change in potential to adjust the COP position within the foot.

Our results suggest that the modulation of COM feedback gains is similar to the modulation of local reflexes during walking. The similarity in phase-dependency and speed-dependency of COM feedback gains and H-reflex during the gait cycle suggests that modulation of COM

gains and local reflexes are related. Both the H-reflex of the soleus and gastrocnemius and the gains of the COM feedback that explains the ankle strategy after perturbations are high during mid-stance [22] and low during early stance, terminal stance and during the swing phase. Furthermore, we also observed that the gains of the COM feedback decrease with increased walking speed and are higher in standing compared to walking, which is similar to the modulation of h-reflex magnitude with walking speed [23].

The sensory mechanism underlying the observed modulation in COM feedback gains during the gait cycle remains unclear. Given changes in foot-ground interaction throughout the gait cycle and the dependency of reflex activity on tactile information from the foot during walking [24, 25], we hypothesized that information from the COP location and vertical ground reaction force could explain the modulation of reflex gains. In contrast with this hypothesis, we found that the fit of the feedback model did not improve when scaling the feedback gains with COP information or the vertical ground reaction force (Fig 9). We believe that there are two potential explanations for this observation. First, this might indicate that tactile information in the foot cannot explain the modulation of COM feedback gains during the gait cycle. Second, this might also indicate that the simplifying assumption of modelling information from cutaneous sensors with linear feedback of vertical ground reaction force or COP location is not valid. In the future, cutaneous stimulations experiments could generate more insight in the modulation of task-level feedback gains during walking [25].

The decrease in $R^2$ values and feedback gains with increased walking speed indicates that task-level feedback of COM kinematics is less important in fast walking. There are multiple explanations for the decreased importance of task-level feedback with increased walking speed. These changes in task-level feedback might reflect an increase in inherent stability of the skeleton system with walking speed. Alternatively, subjects might rely less on an ankle strategy and more on a stepping strategy at higher walking speeds. The ankle strategy can act sooner than the stepping strategy, which has no effect before the next foot contact, might be less important during fast walking [12]. Step frequency increases with increased walking speed, thereby reducing the duration of the swing phase. As a result, the next foot placement occurs earlier and therefore the delay between perturbation onset and foot placement to control balance decreases. In addition, the shorter duration of the stance phase with increased walking speed might limit the potential of the ankle strategy to control balance.

When comparing the different types of perturbations in walking (Fig 10), we found that both the position and velocity feedback gains are higher in the pelvis push perturbations (walking at 1.25 m/s) compared to the support surface perturbations (walking at 1.2 m/s, Fig 10). This might be the result of oversimplification of task-level feedback with linear feedback of COM kinematics. We could only explain up to 80% of the variance in ankle moment with this simple feedback model. Non-linear feedback or feedback from other task-level variables such as COM accelerations or vestibular information [6, 30] might contribute to reactive muscle activity and joint moments in perturbed walking.

### 3.3 Task-level versus joint-level feedback

The observation that reactive muscle activity and joint moments were better described by a model with delayed feedback of COM kinematics, compared to delayed feedback of ankle kinematics, does not mean that local reflexes are not important for human balance control. We only showed that our model, based on linear feedback from ankle angle and velocity cannot explain the reactive muscle activity and joint moments respectively 190ms and 230ms after perturbation onset. The reason for this may be twofold: First, alternative local feedback models, for example based on feedback from muscle fiber kinematics or force, might improve the

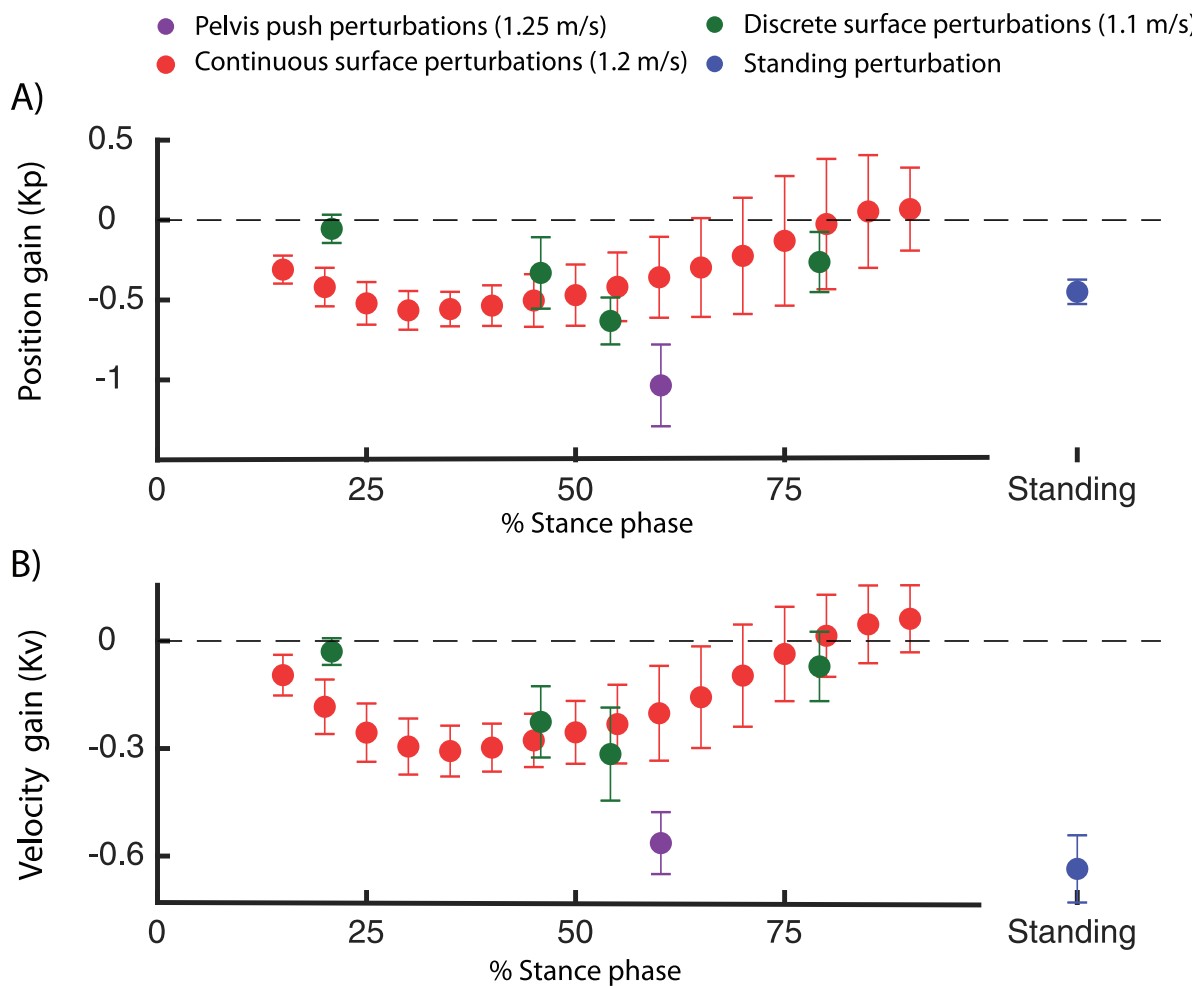

**Fig 10. Position and velocity gains computed in the dataset with perturbed standing (green [26]), walking with discrete (blue [10]) and continuous(red [27]) surface perturbations and discrete pelvis push perturbations (purple [9]).** The feedback gains for walking are presented as a function of the percentage of the stance phase. Note that the percentage in the stance phase represents the phase at which the feedback model was evaluated (i.e. 250 ms after perturbation onset).

variance explained by the local feedback model. Second, 200ms after perturbation onset, the reactive muscle activity observed in the EMG signals and joint moments could be the result from both local reflexes and long latency reflexes in response to the ongoing perturbations. The high variance explained by the COM kinematics feedback model indeed suggests that the response, at the time instance we evaluated it, does not merely result from local reflexes. Hence, our results indicate that, similar as in standing balance [3], long latency responses reflect delayed feedback of task-level variables and not linear feedback of joint kinematics errors.

### 3.4 Contribution of reactive muscle activity

The relation between COM kinematics and reactive muscle activity shows that the reactive ankle joint moments do not solely result from visco-elastic properties of active muscles, but result from active feedback control (Fig 4). Based on the relation between joint moments and delayed COM kinematics, it is unclear if the modulation of ankle moment is indeed an active

control mechanism. Given that a similar correlation was observed at the level of reactive activity of the calf muscles confirmed the important contribution of active feedback control.

Despite the clear contribution of active control mechanisms, it is hard to identify the relative importance of active feedback control and passive dynamic contributions to the ankle moment. Musculoskeletal modelling could be used to identify muscle stiffness and damping using muscle models [31]. This would enable estimation of joint stiffness and damping. However, this relies on accurate methods to estimate the muscle state from motion capture data (e.g. [32]) and on accurate models of muscle contraction dynamics [33].

### 3.5 Derive neuromechanical models for balance control during walking

Our findings have important implications for modelling neuromechanics of walking. In current models, balance is mainly controlled by foot placement [15, 34]. Here, we showed that subjects also adjust the ankle joint moment in response to perturbation during walking and that this strategy can be modelled using delayed COM position and velocity feedback. Hence, we propose an additional supra-spinal feedback loop to model the ankle strategy during walking, similar to the neuromechanical models of perturbed standing [29]. Since COM kinematics feedback largely captures reactive ankle joint moment in response to different types of balance perturbations across walking speeds, we believe that this is an important extension to current state-of-the art neuromechanical models. One of the future challenges is to investigate how this additional COM-feedback loop interacts with local feedback control, for example positive force feedback of the calf muscles as modeled by [35]. We believe that a combination of local and supra-spinal feedback to control the ankle muscles is especially useful in the control of wearable robotic devices. We hypothesize that an additional supra-spinal feedback loop will facilitate shared control of balance when using neuromechanical models in the control of wearable robotic devices.

## 4 Materials and methods

We used existing data from multiple perturbation experiments in healthy young and older subjects to derive sensorimotor transformations for the ankle strategy during standing and walking (see Fig 1 for an overview).

### 4.1 Experimental methods

Integrated 3D motion capture was used in four datasets to measure the human response to perturbations during standing and walking. An overview of the different datasets and measurements is given in Fig 1.

- Standing balance was perturbed using support surface translations triggered at a discrete time instance. Integrated motion capture data was collected in 8 young (age 21 ± 2std) and 10 older adults (age 67 ± 3std). The perturbation protocol and data collection is extensively described in [26]. In short, standing balance was perturbed by means of a sudden forward and backward platform translation with respectively fast, medium and slow acceleration profiles (Fig 1), as well as medio-lateral platform translations, but only forward and backward perturbations were analyzed in this study. Perturbations were applied in a random order. Whole body motion was recorded using 3D motion capture with an extended plug in gait marker (Vicon, Oxford Metrics). Ground reaction forces were collected from two AMTI force plates (AMI, Watertown, USA) embedded in the platform. This dataset has already been used to analyse kinematic strategies in response to backward surface translations in [17]. Here, we additionally analyzed the response to forward support surface translations

and reactive muscle activity of the tibialis anterior, soleus and gastrocnemius, which was collected using surface electromyography (Cometa).

- Steady state walking was also perturbed by means of an external force (push) at the pelvis. Integrated motion capture data was collected in 9 subjects (age 25 ± 2 std years). The perturbation protocol and data collection is extensively described in [9]. In short, the perturbations consisted of a forward of backward directed push at the pelvis with four different magnitudes. Forward and backward push perturbations were applied in a random while the subjects walked at 0.62 m/s and 1.25 m/s. Whole body motion was recorded using 3D motion capture with an full-body marker protocol (VisualEyez-II). Ground reaction forces were collected on a split-belt treadmill (MotekForceLink). Muscle activity of the tibialis anterior and gastrocnemius was measured using electromyography (Delsys).

- Steady state walking was perturbed by means of a sudden change in speed of the treadmill belt, triggered at specific phases of the gait cycle. Integrated motion capture data was collected in 18 young (age 21 ± 2 std years) and ten older adults (age 71 ± 4 years). The perturbation protocol and data collection in extensively described in [10]. In short, the perturbations consisted of a sudden increase or decrease in speed of the treadmill belt with two different magnitudes. These perturbations were applied at four different phases of the gait cycle. Medio-lateral treadmill translations were also part of the perturbation protocol but were not analysed in this study. All perturbations were applied in a random order. All subjects walked at 1.1 m/s on a split-belt treadmill. Whole body motion was recorded using 3D motion capture with an extended plug in gait marker set (Vicon, Oxford Metrics). Ground reaction forces were collected on a split-belt treadmill (MotekForceLink). Muscle activity of the gastrocnemius, soleus, tibialis anterior was measured using surface electromyography (Bortec Octopus 8 channel electromyography).

- Steady state walking was also perturbed by means of continuous changes in the speed of the treadmill belts. Integrated motion capture data was collected in 15 young subjects (age 24 ± 4 std years). The perturbation protocol and data collection is extensively described in [27]. In short, balance was perturbed continuously using three pseudo-random belt speed control signals, with mean velocities of 0.8 $m/s$, 1.2 $m/s$ and 1.6 $m/s$. Each trial with continuous perturbations consists of 480 seconds walking, resulting in approximately 500 gait cycles of perturbed walking for each subject and each walking speed. Whole body motion was recorded using 3D motion capture with an full-body marker protocol. Ground reaction forces were collected on a split-belt treadmill (MotekForceLink). No electromyography data was collected in this study.

## 4.2 Inverse kinematic and dynamic analysis of the experimental data

Joint kinematics and kinetics were computed using a scaled generic musculoskeletal model with 23 degrees of freedom (gait 2392) in OpenSim [36]. This model was scaled to the anthropometry and mass of the subject based on the marker positions and ground reaction forces in the static trials. Joint kinematics of the scaled model were computed from the recorded marker trajectories using a Kalman smoothing algorithm [37]. Joint kinetics were computed based on the equations of motion of the model with OpenSim's inverse dynamics tool. The motion capture dataset with continuous perturbations was processed using the open source software to compute joint kinematics and kinetics provided with the manuscript [27]. Whole body COM position and velocity was computed from the joint kinematics and the model of the skeleton (i.e. kinematic approach), and was expressed relative to the position and velocity of the base of support (i.e. foot in contact with the ground).

## 4.3 Normalization of data

Joint kinematics, kinetics and muscle activity data were normalized to facilitate comparison between subjects and datasets to enable estimation of feedback gains based on data pooled over all subjects. Similar as in [28], COM positions were normalized by $l_{max}$, speeds by $\sqrt{gl_{max}}$, torques by $mgl_{max}$ and muscle activity by MVC values for the standing data and by the peak values of the gait-cycle-average activity observed during unperturbed walking for the walking data. We noticed that normalizing the EMG data was not sufficient to have a reliably comparison of muscle activity between subjects, which might be related to the lack of MVC data the in walking experiments [38]. We therefore fitted the feedback model only on EMG data of individual subjects, and not on the data pooled over all subjects.

## 4.4 Linear regression

We used least squares regression to obtain linear models between the deviations in ankle and COM kinematics (inputs) and reactive ankle joint moment and muscle activity (outputs) after perturbation. We first computed the mean of the inputs and outputs in the unperturbed standing and walking data. Subsequently, we computed the deviation from the means in the perturbed standing and walking data. For example, in standing balance, we first used the data at 0.5 seconds before perturbation onset, averaged over the multiple perturbation trials, to determine the mean of the unperturbed COM kinematics, joint moments and muscle activity. We then computed the deviation from this mean in each perturbed and unperturbed trial. These deviations were used as input for the least squares regression. The same method was used in perturbed walking, but the inputs and outputs were expressed as a percentage of the stance and swing phase. The linear regression was evaluated at 150ms after perturbation onset in all datasets because a large deviation in COM kinematics was observed in all datasets. For the continuous treadmill perturbations [27], we divided the stance duration into 16 bins and computed a linear model for the ankle moment at each of those phases, all with delayed COM kinematics as inputs.

To evaluate the relation between inputs, i.e., COM or ankle kinematics and the ankle joint moment, we performed the linear regression independent of the perturbation direction (i.e. one single regression for both perturbation directions). This decision was mainly based on the observation that the variance explained by a single regression model was similar to the variance explained by separate regression models for each perturbation direction (Fig 4). In contrast, we found that separate feedback gains for the forward and backward directed perturbations were needed to explain the reactive muscle activity (Figs 3H–3J and 4H–4J).

For each dataset, we pooled the data for the different perturbation magnitudes, perturbation direction (forward and backward), repetitions (i.e. trials) to perform a single least squares regression. The uncentered coefficient of determination ($R^2$) (Eq 7) and root mean square error (RMSE) between the measured and reconstructed joint moments and muscle activity are reported to quantify the fit of the linear regression (Eqs 1 and 2).

$$R^2 = 1 - \frac{\sum \left( y_{meas} - y_{mod} \right)^2}{\sum y_{meas}^2} \tag{7}$$

with $y_{meas}$ the measured ankle moments or muscle activity and $y_{mod}$ the reconstructed ankle moments and muscle activity with delayed linear feedback of COM kinematics.

### 4.5 Gains normalized by COP position or vertical ground reaction force

The hypothesis that gain modulation might be driven by cutaneous information from the foot was evaluated by fitting one model with constant feedback gains through all data points of the continuous perturbations (i.e. combing the different gait phases). We computed the $R^2$ and *RMSE* when the feedback gains were proportional to the vertical ground reaction force (Eq 5) or by the minimal distance between the COP and the bounds of the foot (Eq 6) in the fore-aft direction. The bounds of the foot were computed based on the motion capture data.

### 4.6 Statistical analysis

We used statistical tests to evaluate (1) if gain modulation might be driven by cutaneous information from the foot and (2) if the fit with the task-level feedback model is better than the joint-level feedback model. To test the first hypothesis, we used a repeated measures ANOVA to evaluate if the *R* value is significantly different in models with constant gains modulated by the vertical ground reaction force or COP position compared to a model with constant gains Table in S1 Table. A Fisher z-transformation was used to transform the correlation coefficient in normally distributed z-scores. Bonferroni correction was used to correct for multiple comparisons in the post-hoc testing. To test the second hypothesis, we also used a paired t-test, with R-values after Fisher z-transformation, to evaluate if the *R* value is significantly different in the model with task-level feedback compared to a model with joint-level feedback. A two-sided confidence interval with an alpha level of 0.05 was used for all statistical tests.

## Supporting information

**S1 Fig. Sensitivity of the time delay.** We evaluated whether the variance in ankle moment explained by the linear regression is sensitive to the time delay. This additional analysis was important to verify that the observed relation between COM kinematics and ankle moment is a feedback control process, and does not simply reflect the coupling due to the dynamics of the skeleton. We did this sensitivity analysis on the pelvis push perturbations and found that the relation between ankle moment and COM kinematics is indeed sensitive to the neural delay. The variance explained by the linear regression was optimal with a physiological plausible delay of 100ms and decreased strongly for delays smaller than 50 ms or larger than 120 ms. Note that time delay in this study was selected based on literature [29] and was not based on this sensitivity analysis.
(EPS)

**S2 Fig. Deviations in COM position and velocity in the different types of perturbations.** To evaluate if the magnitude of the perturbations are similar between datasets, we compared the deviation in COM position and velocity at 150ms after perturbation onset for the four datasets. We found that the deviation in COM position and velocity is similar in the different datasets of perturbed walking (B-D) and is larger in perturbed walking compared to perturbed standing (A). A similar deviation in COM kinematics is observed when analysing the data 100ms (E), 150ms (F) and 200ms (G) after perturbation onset in the pelvis push perturbations.
(TIF)

**S1 Table. Sensitivity of the time delay.** We evaluated whether the variance in ankle moment explained by the linear regression is sensitive to the time delay. This additional analysis was important to verify that the observed relation between COM kinematics and ankle moment is a feedback control process, and does not simply reflect the coupling due to the dynamics of the skeleton. We did this sensitivity analysis on the pelvis push perturbations and found that the relation between ankle moment and COM kinematics is indeed sensitive to the neural delay.

The variance explained by the linear regression was optimal with a physiological plausible delay of 100ms and decreased strongly for delays smaller than 50 ms or larger than 120 ms. Note that time delay in this study was selected based on literature [29] and was not based on this sensitivity analysis.
(PDF)

**S1 File. DataBeltPerturb.mat.** Matlab file with raw data of the experiment with discrete perturbations in speed of the treadmill belts [10].
(MAT)

**S2 File. DataContinuousPerturb.mat.** Matlab file with raw data of the experiment with continuous perturbations in speed of the treadmill belts [27].
(MAT)

**S3 File. DataPelvisPush.mat.** Matlab file with raw data of the experiment with pelvis push perturbations [9].
(MAT)

**S4 File. DataStandingBalance.mat.** Matlab file with raw data of the experiment with support surface translations during standing [26].
(MAT)

**S5 File. ExamplePlotFigureAfschrift2019.m.** Matlab script to plot the relation between COM kinematics and changes in ankle moment in the discrete belt speed perturbations.
(M)

**S6 File. ExamplePlotFigureVlutters2018.m.** Matlab script to plot the relation between COM kinematics and changes in ankle moment in the pelvis push perturbations.
(M)

**S7 File. ExamplePlotFigureMoore2014.m.** Matlab script to plot the relation between COM kinematics and changes in ankle moment in the continuous belt speed perturbations.
(M)

## Acknowledgments

The authors would like to thank the authors of the public available datasets for sharing their data. In addition, we would like to thank Herman van der Kooij and Edwin van Asseldonk for their insightful discussion related to the modulation of COM feedback during the gait cycle and if this modulation can be explained by altered input from tactile sensors.

## Author Contributions

**Conceptualization:** Maarten Afschrift, Friedl De Groote, Ilse Jonkers.

**Formal analysis:** Maarten Afschrift.

**Funding acquisition:** Maarten Afschrift, Friedl De Groote, Ilse Jonkers.

**Investigation:** Maarten Afschrift, Friedl De Groote, Ilse Jonkers.

**Methodology:** Maarten Afschrift, Friedl De Groote.

**Project administration:** Maarten Afschrift, Friedl De Groote, Ilse Jonkers.

**Software:** Maarten Afschrift.

**Supervision:** Friedl De Groote, Ilse Jonkers.

**Visualization:** Maarten Afschrift.

**Writing – original draft:** Maarten Afschrift, Friedl De Groote, Ilse Jonkers.

**Writing – review & editing:** Maarten Afschrift, Friedl De Groote, Ilse Jonkers.

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
