## [Decision Letter · Decision Letter 0]

17 Dec 2020

Dear Dr. Afschrift,

Thank you very much for submitting your manuscript "Similar sensorimotor transformations control balance during standing and walking" for consideration at PLOS Computational Biology.

As with all papers reviewed by the journal, your manuscript was reviewed by members of the editorial board and by several independent reviewers. I apologize for the lengthy review period for this paper. This was due to waiting for comments from a reviewer who was ultimately unable to complete their reviews. Nevertheless, the remaining two reviewers were quite positive about the manuscript. They did, however, raise several important points related to the logic of the conclusions and presentation of the methods and results. In light of the reviews (below this email), we would like to invite the resubmission of a significantly-revised version that takes into account the reviewers' comments.

We cannot make any decision about publication until we have seen the revised manuscript and your response to the reviewers' comments. Your revised manuscript is also likely to be sent to reviewers for further evaluation.

Sincerely,

Adrian M Haith

Associate Editor

PLOS Computational Biology

Wolfgang Einhäuser

Deputy Editor

PLOS Computational Biology

Reviewer's Responses to Questions

**Comments to the Authors:**

Reviewer #1: Review: PCOMPBIOL-D-20-01750

Title: Similar sensorimotor transformations control balance during standing and walking

This manuscript describes a series of analyses on existing datasets to study how balance is controlled during standing and walking. The authors use these analyses to show that during walking, as during standing, COM state feedback (rather than ankle feedback) is used to control ankle moments (and ankle muscle activities). Overall, this is a nice manuscript, that brings together datasets from several previous datasets to come to an interesting result. I have one somewhat more major comment, and several smaller minor comments.

One of the things I’m kind of left confused about is the role of simple stretch reflexes in all of this; we know they exist (see all the work on H-reflexes that the authors also mention), and that they should have a more or less 1-to-1 mapping between ankle state and muscle activity. Yet, the data of the authors shows that these models fit less well, which, in a way, is surprising to me. Probably, this is due to the fact that the models that are fitted here do not only contain information on short-latency reflexive activity, but also on longer latency mechanisms. Or is there something else that I am missing here? These kind of thoughts could be expressed somewhat more clear in the manuscript.

Minor comments:

1) Figure 2; what are the green lines in this (and other) figures? It seems as if they are fits to just the unperturbed data for a, but for b, it seems for all data (or is that due to the multiple regression that was performed? Also, I was wondering why for the standing, only the COM feedback model is shown, and not the data for the ankle feedback model?

2) Figure 3; from the numbers, the RMSE in G is twice that of D, but if I would have to guess, I would say it’s much larger? (or is this just my eyes being off?

3) It may be nice to show some example traces for one perturbation of COM deviations, and then ankle moment deviations, and estimated ankle moment deviations, just for the reader to get some idea of what’s going on.

4) Figure 4. In figures 2-3 you used arrows with arrowheads pointing toward the feature, and at the tail the thing it indicated. Here, you do the opposite (super-minor, but may be nice to be consistent in this).

5) Page 6, line 183; here, it is stated that the perturbations happened around toe-off, yet the figure that is refereed to has perturbations in many phases of the gait cycle?

6) Figures 6 and 7; I think the figures miss a delta in both the x and y labels, as these are deviations from the nominal (pattern/ activity/etc), right?

7) Page 9, line 277, one of our recent preprints may be of interest here; basically, we showed phase dependent contributions of vestibular info, which also was context dependent. See https://www.biorxiv.org/content/10.1101/2020.09.30.319434v1

8) Page 10, line 234; references that subjects might rely more on stepping (at least in the ML plane) are:

Stimpson, K.H., et al., Effects of walking speed on the step-by-step control of step width. J Biomech, 2018. 68: p. 78-83.

And

https://www.biorxiv.org/content/10.1101/2020.06.10.143867v1

9) Methods; R^2 values are typically not normally distributed, and some sort of a (modified) fisher r-to-z transform is needed before averaging them, or doing stats on them.

Reviewer #2: The manuscript evaluates a linear feedback control models for balance control during standing and walking. The authors compared task-level feedback models that use COM position and velocity changes to control ankle moment or muscles and local feedback models that use ankle angle and angular velocity changes. In all four (one standing and three walking) experiments evaluated in the study, the task-level feedback models showed higher correlation and smaller errors with human data than the local feedback model did. The author conclude that ankle responses to disturbances during standing and walking can be explained by talk-level feedback but not by local feedback.

Strengths:

- The study shows that an ankle control strategy that has been proposed to explain standing balance can also be applied for walking to some extent.

- It is shown that explain standing and walking balance control is not trivial by evaluating a local feedback control model that fails to explain.

- The models are evaluated with multiple (independently conducted) human experiments.

- The findings (that a standing balance control model also can explain balance control during walking) have interesting and important implications for human walking control.

Weaknesses:

- The manuscript lacks the nuanced presentation that, in my opinion, is necessary for such modeling studies. For instance, it is stated in the abstract that “We found that delayed feedback of center of mass position and velocity, but not local feedback of joint positions and velocities, can explain…” and in the results section that “… we showed that … can largely be explained by task-level feedback of COM kinematics and not with local feedback of joint kinematics” However, the study does not show that ankle responses cannot be explained by local feedback; it shows that the local feedback model the authors selected cannot, and it is still possible that a different local feedback model could explain the ankle responses observed in humans. I think this is a critical limitation of the current manuscript and suggest the following modifications:

1) Do not say “local feedback cannot …” as your conclusion. Instead, say “the local feedback model cannot …”

2) In the introduction, discuss the nature of modeling studies. Again, showing that a local feedback model you selected cannot fit/correlate/etc. observed data does not show that local feedback cannot explain observed data.

3) You could/should introduce and discuss that there can be and there are other (task-based and) local models than the ones you selected. For instance, [Song & Geyer, A neural circuitry that emphasizes spinal feedback generates diverse behaviours of human locomotion. The Journal of physiology, 2015] shows that local force feedback for ankle control can contribute to producing walking and maintaining balance. While the model includes a foot placement control (as you cited as “balance is mainly controlled by foot placement […,33]”) it also has local ankle control.

4) In the introduction, you should also justify the models you selected/proposed in your study. I think the linear models you selected are justifiable as it is simple to be considered as a baseline model and have been used in previous studies on standing balance. (In that, if the models have been used in or have been adapted from previous studies, you should clearly state and cite them in the introduction and also when presenting eq (1)~(4).

5) The discussion can be stronger by covering relevant musculoskeletal modeling studies as (there are not too many and) can be suitable for more nuanced discussions:

5-1) Section 3.2 can include the discussion of “the changes in muscle responses do not necessarily indicate modulation of reflex gains” as stated in [33], as the changes in the responses could result from the musculoskeletal configuration/dynamics. I think this is worth emphasizing to point out the limitation of your studies using simple linear models that do not incorporate the complex musculoskeletal dynamics.

5-2) Section 3.4 can mention the potential contribution of feedforward control in balancing as suggested in [Haeufle, Schmortte, Geyer, Müller & Schmitt, The benefit of combining neuronal feedback and feed-forward control for robustness in step down perturbations of simulated human walking depends on the muscle function. Frontiers in computational neuroscience, 2018].

- The presentation is not in its best form. Please find my comments below for more details.

Detailed Comments:

Line 52~53: It is written that the ankle strategy is the “dominant” strategy in the sagittal balance control. However, the following content does not seem enough to support that statement. Either rephrase the sentence, elaborate on what you mean by “dominant”, or add evidence that the ankle strategy is dominant over the stepping strategy.

Lines 89~91: “We therefore hypothesize that tactile sensors also modulate task-level feedback gains and thereby contribute to the observed modulation of reactive muscle activity.” -> Consider rephrasing the sentence so that this hypothesis is secondary to your main hypothesis of task-level vs. local feedback.

Lines 136~137: “Inputs (COM kinematics or ankle kinematics) and outputs (ankle moment) …” -> I suggest you rephrase the sentence so that it holds the main points without the parts in parentheses. In other words, “ankle moment” should not be in parentheses.

The order of the experiments in Fig. 1, 2 Results, and 4.1 Experimental methods are not consistent, thus confusing.

Fig. 1, Continuous translation walking: Is the y-axis treadmill velocity or Com velocity? Please clarify.

Lines 155~156: “reactive ankle joint moments and muscle activity …” -> “reactive ankle joint moments and muscle activity during standing…”

Please explain what you mean by “limitations related to normalizing EMG data” in section 4.3.

Fig 3. Subplot labels are missing A) and are not consistent with the caption.

I would recommend presenting Fig. 2~4 in a consistent manner. For instance, include joint feedback data in Fig 2, and combine Fig 3 and 4. Fig 2 and the combined Fig3+4 could have the same layout.

Table 2. RMS of TA is smaller for the Joint feedback model. p-RMS of 0.004 means that the Joint feedback model is significantly better? Please clarify.

Line 182~186: Is this paragraph only talking about the data around 60 %stride in Fig 5? If so, please clarify it by, for example, marking the 60 % stride data with shaded boxes and referring it to it as “(shaded area in Fig. 5)”. If I am misunderstanding this paragraph, then please clarify what you mean by “(also applied around toe-off of the contralateral leg)” at line 183.

Fig. 5: Are the data for one speed?

Lines 195~196: Indicate what gait phases you are indicating to by initial, mid-, and terminal stance. E.g. terminal stance: 62.5%, etc.

Fig 6, Row 2~4: Why are the regressions done together for forward and background perturbations? Or, why are they done separately for data in Fig 2 and 4?

Fig 6&7: Are the x-axes COM or Del_COM? If they are COM please explain.

Why is there no Table that summarizes the R^2 and RMS values for the continuous speed perturbation experiment?

Line 242: “(Fig. 9)” -> “(Fig. 9-F)”

Line 245: “(Fig. 9)” -> “(Fig. 9-B and F)” or remove

Line 259: “(Fig. 9)” -> “(Fig. 9-C, D, G, and H)”

Please check the writing/editorial mistakes throughout the manuscript. A few examples are:

- Line 39: “are less well studied” -> “are less studied” or “has received less …”

- Line 54: “fore-after” -> “fore-aft”

- Line 59: “Is is” -> “It is”

- Lines 201~202: “(one representative subject: figure 6A, all subjects figure: 5)” -> “(Fig. 5: all subjects, Fig 6: one representative subject)”

- Fig. 6, Row 1: Mark another value in the y-axis to show the scale.

- Fig. 7: Indicate % stance/stride as in Fig 6.

- Fig. 9: Place the subfigure labels and y-axis labels (e.g., D and H) to not invade the other subfigures.

- Line 326: “pcontact” -> “contact”

- “Fig 11.” -> “Fig. S1”

- “Fig 12.” -> “Fig. S12”

**Have all data underlying the figures and results presented in the manuscript been provided?**

Reviewer #1: Yes

Reviewer #2: Yes

PLOS authors have the option to publish the peer review history of their article (what does this mean?). If published, this will include your full peer review and any attached files.

Reviewer #1: **Yes: **Sjoerd M. Bruijn

Reviewer #2: No
---

## [Decision Letter · Decision Letter 1]

19 Mar 2021

Dear Dr. Afschrift,

Thank you very much for submitting your manuscript "Similar sensorimotor transformations control balance during standing and walking" for consideration at PLOS Computational Biology. As with all papers reviewed by the journal, your manuscript was reviewed by members of the editorial board and by several independent reviewers. The reviewers were mostly satisfied with the revisions to the paper. However, Reviewer 2 raised some outstanding concerns related to presentation. Based on the reviews, we are likely to accept this manuscript for publication. But first please consider the criticisms and suggestions from Reviewer 2 as you may find their comments instructive for further strengthening the paper. I leave it to your judgement, however, as to the extent of the revisions that you wish to make before resubmitting.

Sincerely,

Adrian M Haith

Associate Editor

PLOS Computational Biology

Wolfgang Einhäuser

Deputy Editor

PLOS Computational Biology

[LINK]

Reviewer's Responses to Questions

**Comments to the Authors:**

Reviewer #1: The authors have done a great job in revising the manuscript.

Reviewer #2: I appreciate the authors’ work in addressing most of my comments. I especially acknowledge the added discussion on other factors that could potentially explain the results of this study.

However, I unfortunately need to point out that the writing is not in its best form. Some parts, especially the introduction, even became more difficult for me to read than the original manuscript. Perhaps this happened in the process of modifying the manuscript to address the reviewers’ comments while not considering the overall flow of the paragraphs. While I will leave a few specific comments below, I strongly recommend the authors to proofread the entire manuscript, especially the introduction section, to improve the clarity and readability.

Lines 30~31: “When standing balance is perturbed, the central nervous system estimates the movement of the whole body center of mass to activate muscles and control balance” This is too assertive in my opinion. Rather suggest strong enough citations or modify to something like “Previous studies suggest that …”

It was very difficult to read the introduction. I needed to stop a lot to understand how some parts make sense in the broader context. I think a major revision of the entire section is necessary, so it is not so easy to suggest specific small changes. Thus, rather than giving specific suggestions, I will list a few specific points I think made the section difficult to read:

- Regarding the overall structure of the introduction, currently, many paragraphs consist of both the background and the specific ideas of this study (e.g., Lines 53~54, Lines 87~90, Line96~98, Lines 110~112). I think jumping back and forth between background and what is done in this study harms the readability. It could be easier if the introduction first covers the relevant background then describe what and how this study would address the questions. Maybe this can be done by rephrasing some sentences. For instance, instead of writing “… we hypothesize …” (Line 96) in the middle of a background paragraph, it could be stated as a general idea (and later in the “In this study, …” paragraph you can pick up on that idea).

- Lines 41~42: “seems to be simple” and “This is remarkable…” do not read naturally together, although I understand what it means. Maybe you can change the first sentence to something like “Humans can stand and walk … without difficulty …”

- Lines 83~85: The sentence does not flow well in the paragraph. You should provide some flow/reasoning why you are making this statement here.

- Line 126~128: I think you should merge this paragraph with the previous one (with a good flow rather than simply connecting them).

I still could spot some typos:

- Line 38: “can be can be” -> “can be”

- Line 377: “season” -> “reason”

**Have all data underlying the figures and results presented in the manuscript been provided?**

Reviewer #1: Yes

Reviewer #2: Yes

PLOS authors have the option to publish the peer review history of their article (what does this mean?). If published, this will include your full peer review and any attached files.

Reviewer #1: No

Reviewer #2: No

Figure Files:

Data Requirements:

Reproducibility:

References:

---

## [Editor Report · Decision Letter 2]

24 May 2021

Dear Dr. Afschrift,

We are pleased to inform you that your manuscript 'Similar sensorimotor transformations control balance during standing and walking' has been provisionally accepted for publication in PLOS Computational Biology.

Best regards,

Adrian M Haith

Associate Editor

PLOS Computational Biology

Wolfgang Einhäuser

Deputy Editor

PLOS Computational Biology

---

## [Editor Report · Acceptance letter]

22 Jun 2021

PCOMPBIOL-D-20-01750R2 

Similar sensorimotor transformations control balance during standing and walking

Dear Dr Afschrift,

I am pleased to inform you that your manuscript has been formally accepted for publication in PLOS Computational Biology. Your manuscript is now with our production department and you will be notified of the publication date in due course.

With kind regards,

Katalin Szabo
